# Implementing a Pulmonary Telerehabilitation Program for Young Adults with Post-COVID-19 Conditions: A Pilot Study

**DOI:** 10.3390/healthcare12181864

**Published:** 2024-09-16

**Authors:** Wilmer Esparza, Alfredo Noboa, Camila Madera, Patricia Acosta-Vargas, Gloria Acosta-Vargas, Mayra Carrión-Toro, Marco Santórum, Manuel Ayala-Chauvin, Guillermo Santillán

**Affiliations:** 1School of Physical Therapy, Pontifical Catholic University of Ecuador, Quito 170143, Ecuador; amadera681@puce.edu.ec; 2School of Physical Therapy, Universidad de Las Américas, Quito 170513, Ecuador; alfredo.noboa@udla.edu.ec (A.N.); guillermo.santillan@udla.edu.ec (G.S.); 3Intelligent and Interactive Systems Lab (SI2 Lab), Universidad de Las Américas, Quito 170513, Ecuador; 4School of Medicine, Pontificia Universidad Católica del Ecuador, Quito 170143, Ecuador; gfacosta@puce.edu.ec; 5Departamento de Informática y Ciencias de la Computación, Escuela Politécnica Nacional, Quito 170143, Ecuador; mayra.carrion@epn.edu.ec (M.C.-T.); marco.santorum@epn.edu.ec (M.S.); 6Centro de Investigación en Ciencias Humanas y de la Educación “CICHE”, Universidad Indoamérica, Quito 170103, Ecuador; mayala5@indoamerica.edu.ec

**Keywords:** telerehabilitation, COVID-19, pulmonary rehabilitation, aerobic exercises, young adults

## Abstract

Background: Several studies have shown that pulmonary telerehabilitation (PTR) improves respiratory capacity. However, there is little evidence of its effectiveness in youth with post-COVID-19 conditions (PCC). This study analyzed the effects of a PTR program on young adults with PCC. Methods: Sixteen youths were randomly assigned to a control group (CG) or an experimental group (EG), with eight participants each. The EG participated in a PTR program that included twelve remote, asynchronous four-week sessions with diaphragmatic breathing and aerobic exercises. Vital signs (SpO_2_, HR, RR, BP), physical capacity (sit-to-stand test), cardiorespiratory capacity (6-Minute Walk Test), and perceived exertion (Borg scale) were assessed in both groups. Results: Statistical analyses showed a significant decrease in RR and HR (*p* < 0.012) and an increase in SpO_2_ (*p* < 0.042), physical (*p* < 0.012), and respiratory (*p* < 0.028) capacity. Perceived effort decreased significantly in both groups (CG: *p* < 0.006; EG: *p* < 0.001) only for physical but not for cardiorespiratory capacity (*p* < 0.106). There were no statistical changes registered in BP (*p* > 0.05). Conclusions: The PTR program, which includes respiratory and aerobic exercises, is feasible and effective in improving physical and cardiorespiratory capacity in young people with PCC, as well as reducing HR, RR, and dyspnea.

## 1. Introduction

Recent studies indicate that 40% to 70% of patients have residual symptoms up to three months post-SARS-CoV-2 infection [1,2]. The PCC is characterized by symptoms persisting for at least two months, typically three months after the onset of COVID-19, without an alternative diagnosis [3]. Common symptoms at follow-up include fatigue, dyspnea, sleep disturbances, difficulty concentrating, exertional intolerance, and myalgia [4]. Fatigue and dyspnea are particularly prevalent, with a pooled prevalence ranging from 27% to 58%.

In young adults (aged 12 to 25 years), PCC prevalence has been reported at 49% during the early convalescent stage and six-month follow-up. Notably, PCC is linked more with the severity of initial symptoms and psychosocial factors than biological markers specific to the viral infection [5]. Among university students diagnosed with COVID-19, frequent symptoms included exercise intolerance, dyspnea, chest pain, chemosensory impairment, and loss of appetite, with a third of this group not seeking medical attention [6]. The absence of medical consultation is likely due to the asymptomatic or mild nature of symptoms in this demographic [7,8].

Multiple studies in acute and subacute phases have shown that PTR significantly improves respiratory capacity [9,10]. Research of moderate quality further supports telerehabilitation as a viable and effective means of continuing care. Specifically, PTR has maintained the quality of rehabilitation while reducing infection risks and eliminating the need for travel [11]. Despite the challenges posed by technical difficulties and the lack of direct contact with healthcare providers, evidence indicates that tailored programs can enhance dyspnea management, health status, and exercise capacity, often matching or surpassing the results of face-to-face PTR [12,13,14]. Moreover, patients with chronic obstructive pulmonary disease have reported high levels of acceptability and satisfaction with PTR [15].

PTR is effective as a non-pharmacological treatment for PCC [16]. It typically includes respiratory and aerobic exercises to improve ventilation mechanics and reduce fatigue. Respiratory exercises (anaerobic/aerobic) and stretching have been used to recover pulmonary function, always considering the patient’s condition and vital signs [17]. Diaphragmatic pattern breathing exercises activate inspiratory flow and volume, favoring oxygenation and airway patency [18]. In contrast, aerobic exercise activates the immune response, increases lung tissue flexibility, improves circulation, and prevents oxidative damage [19]. 

Recent recommendations suggest incorporating respiratory physiotherapy and aerobic exercises in PTR to enhance cardiorespiratory fitness in PCC patients [20]. However, most have focused on older adults, leaving a gap in understanding the effects of PTR on young individuals. This study aims to analyze the impact of a PTR program on young adults with PCC. We hypothesized that the program could positively modify vital parameters, reduce dyspnea, and improve physical and cardiorespiratory capacity. 

## 2. Materials and Methods

### 2.1. Setting

This longitudinal retrospective study was conducted at the Universidad de Las Américas “UDLA” in Quito, Ecuador. The participant recruitment period began in January 2023 and ended in March 2023. The university’s occupational physician assessed students diagnosed with PCC for inclusion in the study.

In this context, exposure refers to participating in a 4-week PTR program designed for this study. Follow-up was conducted during the intervention period, from April to May 2023, with clinical assessments performed at the beginning and end of the program.

Data collection occurred from April to June 2023, encompassing the initial clinical data collection and post-intervention evaluation. All data were collected and managed at the university’s Centro de Atención y Rehabilitación en Fisioterapia (CARF), ensuring consistency and accuracy in measuring clinical outcomes.

### 2.2. Design

A longitudinal, retrospective study was conducted on adults (>18 years). The clinical practical protocol respected the stipulated in the Declaration of Helsinki. All participants signed an informed consent form accepting the treatment. The treatment included the techniques and exercises used in the current clinical practice of pulmonary rehabilitation [9,10].

### 2.3. Participants

Participants were students diagnosed with PCC by the occupational physician at the university. The nose swab PCR test for COVID-19 was used to diagnose COVID-19. Medical files that included (i) post-COVID-19 in the acute phase, (ii) previous pulmonary diseases, (iii) comorbidities, (iv) oxygen-dependent, and (v) participation in a PTR before the study were excluded. The medical files obtained were randomly allocated into two groups (CG and EG). The CG has not received any treatment, while the EG follows the PTR program.

### 2.4. Assessments

Participants underwent a complete clinical examination by the university’s Occupational Health Department at the program’s beginning and end. The assessments were performed at the CARF by a physical therapist trained in pulmonary rehabilitation. CARF is a university physiotherapy center that provides specialized care to the community.

The parameters assessed were: (i) vital signs such as oxygen saturation (SpO_2_), heart rate (HR), respiratory rate (RR), and blood pressure (BP); (ii) physical capacity; (iii) cardiorespiratory capacity; and (iv) perceived effort “dyspnea/fatigue.” A finger pulse oximeter (Finger Oximeter model XY-010, manufactured in Italy by GIMA 2021, Gessate (MI)) was used to measure SpO_2_ and HR. 

Physical capacity was assessed by the number of times the subject could stand up and sit down as quickly as possible for 1 min (1 min sit-to-stand test). The test should be performed from a 46 cm highchair without armrests. The participant’s vital signs are taken before and after the test; normal ranges are 30 to 40 repetitions/minute [21]. 

The 6-Minute Walk Test (6MWT) was used to evaluate cardiorespiratory capacity, using meters as the unit of measurement. This test predicts through an equation the distance a person should walk, considering age and weight [22]. Finally, the Borg scale (0–10) was used to measure the perceived exertion during the 1 min sit-to-stand test and the 6MWT. This scale subjectively assesses effort (0 = no effort and 10 = extreme effort) [23]. All participants were assessed at two-time points. The EG was evaluated before and after the intervention, while the CG was assessed simultaneously as the EG.

### 2.5. Intervention

A four-week PTR program was developed to help the participants continue their home rehabilitation program. Participants were required to perform three PTR sessions per week until 12 sessions were completed. Each PTR session was carried out remotely and asynchronously. 

The PTR program was developed on a web page for a follow-up guide to explain the correct techniques for performing the exercises included in the PTR program. The participants of EG had to connect to a web page designed specifically for this CARF project between the two assessments. The home page contains user instructions, and the therapy program is spread over three links (https://rehabilitacionpostcovid.wordpress.com/ accessed on 25 April 2024). In the first link, the instructions for executing the breathing exercises are presented; the aerobic exercises are in the second; and a combination of the previous exercises appears in the third. Each link describes the starting position, execution, number of sets, repetitions, and a video demonstration for each exercise. The use of both types of exercises is expected every day in clinical pulmonary rehabilitation practice.

The EG started the program simultaneously, which allowed the links to be activated progressively as the treatment progressed. Selective activation prevented performing exercises that did not correspond to that week. Participants had to log in to the website and record the end of the rehabilitation program session to proceed with the next session.

The program included diaphragmatic pattern breathing exercises (4 sets of 10 repetitions, with a 1 min pause between each set) and active cycling (4 sets of 6 repetitions, with a 40 s pause between each set), as well as aerobic and/or strength exercises (10 exercises, see Table 1). Respiratory exercises were performed to re-educate the physiological breathing pattern, which increases lung volumes and oxygenation and decreases respiratory rate [24]. Aerobic exercises were chosen because they activate abdominal muscle synergy, increasing lung expansibility [19]. The PTR program was considered complete when students finished the program (12 sessions).

### 2.6. Data Analysis

The statistical analysis was performed with SPSS, version 25.0 (IBM Corporation, Armonk, NY, USA). Quantitative variables were presented as mean (M) and standard deviation (SD). The effect size was calculated using Eta squared (η^2^), where values of 0.01, 0.06, and 0.14 represent small, medium, and large effect sizes, respectively [25]. Qualitative variables were reported as frequencies (N) and percentages (%). The Shapiro–Wilk test showed that the data set was usually distributed for the quantitative variables. An ANOVA analysis was employed (2 groups × 2 measures) to compare mean differences between groups. Tukey’s post hoc test was applied when an interaction between variables was found. The student test was used to compare the age and persistent symptoms between the groups. Categorical data were analyzed using the chi-square test. A value of *p* < 0.05 was considered statistically significant.

## 3. Results

The retrospective study included thirty-seven students’ medical files; twenty-one were withdrawn. The selection of subjects is presented in the flow chart (Figure 1). 

The flow chart shows that of the 37 participants initially assessed, only 18 were eligible and randomly assigned to the two groups (intervention and control). Throughout the study, both groups retained most participants until the end, with only one participant discontinuing the intervention. Finally, 16 participants (8 in each group) were analyzed, indicating that the study had good follow-up and retention of participants, although with small losses during the process.

Concerning the clinical and demographic characteristics, there were no statistically significant differences between patient groups for age, sex, most important sequelae, and duration of symptom persistence after overcoming COVID-19. There were no significant differences between and within the groups for the sex variable (*p* > 0.05) (Table 2). 

The ANOVA analysis (Groups X Measures) did not reveal a significant interaction (F(1.14) = 2.4128, *p* > 0.143, η^2^ = 0.08) for the SpO_2_ variable, nor for the blood pressure variable (F(1.14) = 1.7527, *p* > 0.207, η^2^ = 0,02). On the contrary, a significant interaction was found for the heart (F(1.14) = 7.9042, *p* < 0.014, η^2^ = 0.08) and respiratory rates (F(1.14) = 18.262, *p* < 0.001, η^2^ = 0.28). Post hoc Tukey analysis showed a significant decrease in heart (*p* < 0.001) and respiratory rates (*p* < 0.031) after treatment in the EG. There were no significant differences between groups in any of the variables measured before treatment (*p* > 0.05) (Table 3).

Regarding the physical and cardiorespiratory capacity, a significant interaction (Group X Measurement) was found for the comparison of physical (F(1.14) = 15.121, *p* < 0.006, η^2^ = 0.03) and cardiorespiratory capacities (F(1.14) = 15.121, *p* < 0.002, η^2^ = 0.27). Tukey’s test showed a significant increase in the number of times the subject sat on a chair for 1 min in the EG after treatment (*p* < 0.002). This considerable increase was also observed in the distance traveled by the EG patients (*p* < 0.001). There was no significant difference between the groups before treatment (*p* > 0.05) (Table 3).

Regarding the perceived effort, the Borg scale showed a significant interaction (Group X Measurement) to assess perceived exertion after performing the 1 min sit-to-stand test (F(1.14) = 10.000, *p* < 0.007, η^2^ = 0.13). The post hoc test showed a significant difference between pre- and post-treatment within the groups (CG: *p* < 0.006; EG: *p* < 0.001, respectively) (Figure 2). Concerning the Borg scale applied before and after, the 6MWT did not show any significant interaction (F(1.14) = 2.9929, *p* > 0.106, η^2^ = 0.11).

## 4. Discussion

This study aimed to analyze the effects of a PTR program on young adults with PCC. The results showed that respiratory and aerobic exercises decrease RR and HR, improving physical and respiratory capacities in this condition.

SpO_2_ increased in both groups at the end of the treatment (CG: 0.7%; EG: 2.3%). SBP and DBP decreased in the EG (11; 2.7 mmHg) and increased in the CG (4.6; 1.8 mmHg). However, none of these changes were significant. This result is consistent with a recent study that continuously measured blood pressure, showing no changes in this parameter in young patients [26]. HR and RR increased in the CG after treatment (HR = 0.9; RR = 1) but decreased significantly in the EG (HR = 4; RR = 8.7). The decrease in these parameters in the EG could have been due to improved voluntary breathing control. Specifically, controlled breathing exercises produce a decrease in RR and HR, as well as an increase in lung volume [27].

Regarding physical capacity (strength) and cardiorespiratory capacity (endurance), both groups significantly increased their performance, but only the EG showed a significant improvement. The EG increased the number of times it could stand up and sit down from a chair by 1.7 and the distance covered after 6 min by 7.5 m. The improvement in physical and cardiorespiratory capacity in the CG may be due to the spontaneous recovery experienced by the young subjects. The EG significantly increased the physical (4.9) and cardiorespiratory (217.5 m) capacity to perform these activities. This increase in physical capacity is consistent with the results of Dalbosco-Salas et al. (2021) [17] in an observational study of 24 PTR treatment sessions. However, our average difference was higher (+3.2 times); this could be due to the age of the subjects (21.7 vs. 55.6 years), as the treatment was similar. In any case, increased cardiorespiratory capacity has been reported in post-COVID-19 patients in the acute phase after respiratory and aerobic exercise [28].

Finally, both groups’ perceived exertion (fatigue) decreased after evaluating physical and respiratory capacities. However, this decrease was statistically significant only after performing the 1 min sit-to-stand test (CG: 1.1; EG: 2.4 points). Probably, the 6MWT is not sufficiently demanding to cause fatigue during exertion, or the respiratory and aerobic exercises improved fatigue tolerance. The 6MWT and sit-to-stand test are generally used as the primary tests to quantify physical and cardiorespiratory fitness. However, a recent study that used a spirometer to quantify respiratory function in isolation found no significant differences between the two groups, although the 6MWT and sit-to-stand test presented significant differences. The authors suggested that the 6MWT and sit-to-stand test provide more information about the cardiovascular response than the respiratory response. Therefore, a spirometer is essential to reliably assess respiratory function in isolation [29].

It is important to emphasize that this study’s results are hypothesis-generating and aim to pave the way for future research. Although improvements in physical and cardiorespiratory capacities have been identified, these should be interpreted cautiously. This study contributes preliminary data about the physical and functional characteristics of young patients with PCC and confirms the suggestion of Silva-Santos et al. [20] to simultaneously implement a pulmonary rehabilitation program that includes pulmonary physiotherapy and aerobic exercises. Our PTR program considered some fundamental aspects in its organization, including (i) being developed for an accessible population (young adults); (ii) autonomy for connection to the platform; (iii) clear instructions for monitoring the program; (iv) integration of aerobic exercises into a pathology that was initially considered solely respiratory; and (v) progression of the proposed respiratory and aerobic exercises. This organization allowed all cardiorespiratory parameters to recover progressively without putting the patient at risk.

Moreover, we can point out that the variety of exercises in this PTR program allows patients to maintain their interest. This adherence to the program means that the training sessions are not perceived as repetitive. Another aspect to highlight is improving the participants’ quality of life. Although this parameter was not evaluated, patients reported feeling more capable of performing their daily activities. This case positively impacted their emotional well-being and the perception of their relatives. Finally, the platform’s design and configuration are user-friendly and easy to access. Further studies recruiting a more significant and representative population are needed to verify our findings.

## 5. Conclusions and Limitations

While this study provides valuable insight into implementing a pulmonary PTR program combining respiratory and aerobic exercises, it is essential to acknowledge several limitations that may influence the generalizability and interpretation of our findings.

First, the small sample size of our study limits statistical power and may impact the robustness of the results. With only 16 participants, the study’s ability to detect smaller effect sizes is limited, and the findings should be considered preliminary. Future studies with larger sample sizes must confirm these results and provide more definitive conclusions.

Second, the specific population studied (young adults enrolled at a university) may not represent the broader population affected by PCC. Participants were generally healthy young adults, which may limit the applicability of these findings to other age groups or people with more severe or chronic conditions. The study’s focus on a relatively homogeneous group also raises questions about how these findings can be generalized to more diverse populations.

Furthermore, the study was conducted in Quito, Ecuador, at 2800 m above sea level. High altitude could have influenced several physiological parameters, such as oxygen saturation (SpO_2_), which is typically lower at higher altitudes. This environmental factor may have affected baseline and post-intervention measurements of SpO_2_ and other respiratory metrics, potentially confounding the results. Therefore, the findings may not directly apply to lower altitude populations.

The socioeconomic status of the participants, who were students at a private university, may also influence the results. Higher socio-economic status is typically associated with better access to healthcare, nutrition, and general living conditions, which could have contributed to the observed improvements. This factor should be considered when interpreting the results, as results may differ in populations with lower socio-economic status.

Given these limitations, it is critical to interpret the results with caution. The improvements in physical and cardiorespiratory capabilities observed in this study are promising but should be considered hypothesis-generating rather than conclusive. The results suggest the potential benefits of a PTR program for young adults with PCC, but further research is needed to validate these findings in more extensive and diverse populations.

While the study demonstrates a PTR program’s feasibility and potential efficacy, it is essential to recognize that these findings are preliminary and should not be overgeneralized. The observed reductions in HR, RR, and dyspnea are encouraging but need to be corroborated by additional studies addressing the abovementioned limitations.

The findings should be considered a foundation for future research rather than final evidence of program effectiveness. We recommend additional studies with more extensive and diverse populations to validate and extend these findings, ensuring that conclusions apply to a broader demographic.

## Figures and Tables

**Figure 1 healthcare-12-01864-f001:**
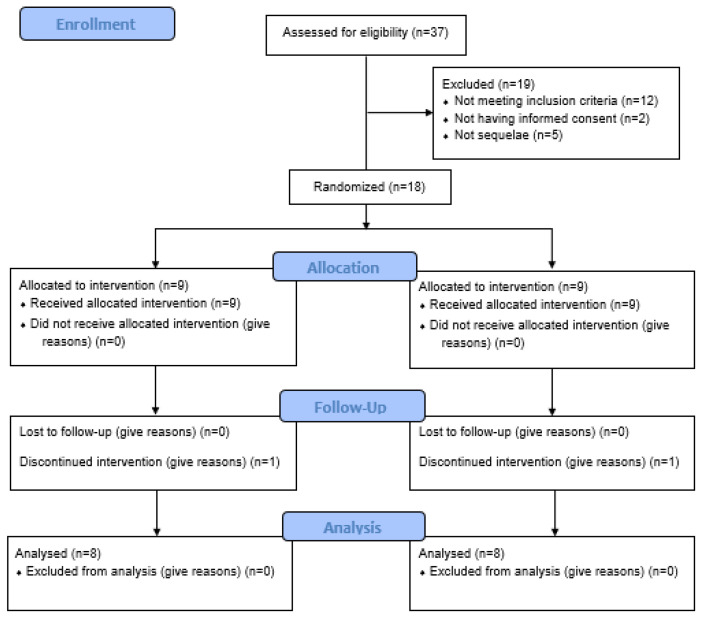
Flow chart.

**Figure 2 healthcare-12-01864-f002:**
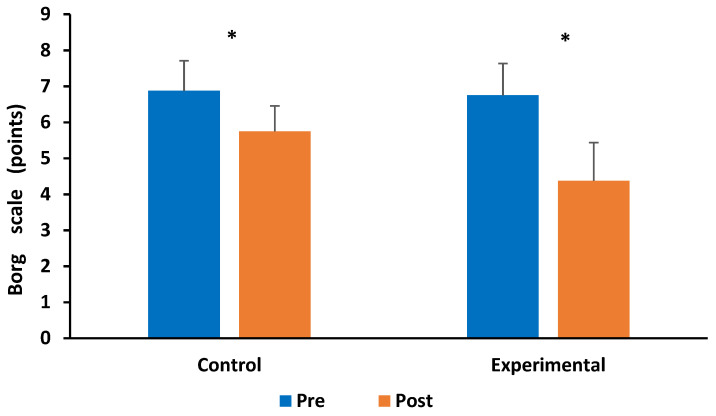
Borg scale results after the 1 min sit-to-stand test. * Significant differences within the groups.

**Table 1 healthcare-12-01864-t001:** Exercises proposed in the program.

Exercise Type	Name	Number of Repeats	Number of Series	Series Breaks	Execution Time
Breathing	Diaphragmatic pattern	10	4	1 min	4 to 8 min
	Active cycle	6	4	40 s	8 to 10 min
	Abdominal hollowing	12	3	1 min	6 to 9 min
Aerobic	Bridge	20	4	40 s	6 to 12 min
	Unilaterally supported bridge	20	4	40 s	
	4-point position	14	5	1 min	
	4-point position with hip dissociation	20	5	1 min	
	High knee extension with raised pelvis and support	20	4	40 s	
	Squat	20	4	30 s	
	Stride	12	4	30 s	
	Tiptoe lift	20	4	30 s	
	Burpee	8	4	30 s	

**Table 2 healthcare-12-01864-t002:** Demographic and clinical characteristics of the groups.

Variables	CG	EG	
	N = 8 (50%)	N = 8 (50%)	*p* Value
**Sex**			
Men	2 (33.33)	4 (66.67)	
Women	6 (60.00)	4 (40.00)	0.301 †
**Sequel**			
Dyspnea	4 (44.44)	5 (55.56)	
Fatigue	4 (57.10)	3 (42.90)	0.614 †
	Mean (SD)	Mean (SD)	
**Age (years)**	21.7 (2.3)	21.1 (1.6)	0.709 *
**Persistence of Symptoms (months)**	2.62 (0.74)	2.25 (0.88)	0.375 *

* Student Test; † Chi-square test.

**Table 3 healthcare-12-01864-t003:** Pre- and post-treatment values for vital signs and physical and cardiorespiratory capacities.

Variable	Control Group			Experimental Group		
	Mean (Sdt)		*p*-Value	Mean (Sdt)		*p*-Value
	Pre	Post		Pre	Post	
**Vital Signs**						
SpO_2_	91.4 (1.7)	92.1 (1.9)	0.401	91.3 (2.6)	94.3 (1.4)	0.042 *
RR	72.8 (11.9)	73.8 (8.7)	0.779	74.1 (7.3)	65.4 (5.3)	0.012 *
HR	20.5 (2.7)	21.4 (2.4)	0.401	21.6 (1.6)	17.6 (1.2)	0.012 *
SBP	107.8 (9.3)	112.4 (11.5)	0.746	122.5 (12.0)	111.5 (8.8)	0.122
DBP	80 (8.7)	81.8 (7.0)	0.484	81.1 (9.1)	78.4 (4.6)	0.208
**Physical capacity**						
6MWT	461.6 (86.8)	469.1 (85.9)	0.084	403.3 (53.9)	620.8 (131.9)	0.028 *
Borg scale	5.4 (1.2)	5.1 (1.2)	0.969	5.8 (0.7)	4.1 (1.0)	0.051
**Cardiorespiratory capacity**						
Sit-to-stand	25.8 (3.6)	27.5 (5.0)	0.059	29 (5.9)	33.9 (5.3)	0.012 *
Borg scale	6.9 (0.8)	5.8 (0.9)	0.006 *	6.8 (0.7)	4.4 (1.1)	0.001 *

SpO_2_ = oxygen saturation; HR = heart rate; RR = respiratory rate; SBP = systolic blood pressure; DBP = d iastolic blood pressure. * Statistically significant.

## Data Availability

The corresponding author can provide data supporting the reported results upon reasonable request.

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
