# Peer review of "Implementing a Pulmonary Telerehabilitation Program for Young Adults with Post-COVID-19 Conditions: A Pilot Study"

_healthcare, 2024, doi:10.3390/healthcare12181864_

Round 1

Reviewer 1 Report

Comments and Suggestions for Authors

The paper by Esparza et al. presents the assessment of a pulmonary telerehabilitation (PTR) program for young adults with post-COVID-19 condition (PCC). In the study, 16 participants were divided into control and experimental groups, and the latter was treated with a 4-week remote PTR program including diaphragmatic breathing and aerobic exercises. The study evaluated vital signs, physical and cardiorespiratory capacity, and perceived effort. The results show that in the experimental group, respiratory rate, heart rate, oxygen saturation, physical capacity, and respiratory capacity presented a significant improvement. Perceived efforts for physical tasks significantly decreased in both groups. The difference in blood pressure did not show significant changes after treatment.

The introduction primarily focuses on detailing the COVID-19 pandemic rather than introducing the topic of pulmonary telerehabilitation (PTR) programs. It would greatly enhance the abstract if the author provided a comprehensive review of the benefits, pros and cons, implementation ease, cost-effectiveness, and accessibility of PTR programs. This would offer readers a clearer understanding of the context and significance of the study within the broader landscape of pulmonary rehabilitation.

The section on sensing and assessment lacks detailed information about the specific tools or methods the author intends to employ. Providing clarity on the chosen sensing technologies and assessment methodologies would enhance the readers' understanding of how data will be collected and analyzed in the study.

It makes the discussion too long and not engaging for the actual results from this research. The discussion should be condensed and tied to relate directly to the results being discussed. This would provide better clarity and retention for the reader.

Major Points:

The young adults with post-COVID-19 conditions benefited well from the Telerehabilitation Program. However, the main limitations associated with the study, such as the small sample size, make the results less strong and generalizable.

The author should include a meta-analysis in the paper.

Is there any difference between males and females? How have the exercises been controlled? And have all participants completed the exercise plan?

Where were the assessments conducted, before and after the intervention?

Figure 2 should indicate the actual data points to accurately show the significance of the results.

As stated by the author, this is a pilot experiment, and it does not fully represent the outcomes of the Telerehabilitation Program.

Minor points:

Few grammatical corrections.

Comments on the Quality of English Language

There are a few grammatical corrections.

Reviewer 2 Report

Comments and Suggestions for Authors

Implementing a Pulmonary Telerehabilitation Program for 2 Young Adults with Post-COVID-19 Condition. A Brief Report

Generally, the manuscript quality needs to be enhanced.

Abstract:

It is not clear if the control group enrolled in a PR program or continued their ADLs as it is. Also reporting the results. I suggest that authors rewrite the abstract to make it clearer to the reader.

Intro:

Line 39, 61: PCC, PTR. Define in full first. Defining in the abstract does not count.

Line 53: Add reference.

On the hand phrase was used few times.

Line 61 and 71. Talk about telemedicine first and then starts with PTR. You set the scene to the reader by that. Currently it is the opposite.

Line 79. This paragraph is not clear. Please rewrite and make it clearer and the ideas are more connected.

Methods:

Line 124: A 4-week PTR program was developed to help the participants to continue your academic program at home.

What is YOUR ACADEMIC? Typo?

Have you looked at the classification of disease severity, and took this into account?

All included participants were only students? Sometimes authors refer to participants as students, and sometimes patients.

Power calculation? based on what? What outcome measure?

Results:

HR value were in 20s?? and it reduced to 17 post in PTR?

The values of HR and RR were inversed

Discussion:

Pilot study?

Authors only mentioned that this study is a pilot study in the discussion?

Comments on the Quality of English Language

As above

Round 2

Reviewer 1 Report

Comments and Suggestions for Authors

Unfortunately, the authors have not updated Figure 2, which fails to show the actual significance because the error bars are almost identical. This is my strongest criticism of the paper and should be easily addressed as per my previous review comment. It is disappointing that this simple data presentation was not performed to remove a clear potential source of biased results.

Additionally, while the authors have added a meta-analysis as a reference, this does not clarify the scope of expectation bias in these data, particularly given the low number of participants. The addition of error significance only allows for assumptions of significance between groups, suggesting the effect of implementing a Pulmonary Telerehabilitation Program may or may not be effective in treating post-COVID-19 syndrome in young patients. One way to address these doubts could be to include the actual data points in the graphic.

Adding more information to the introduction does not help to make the brief report more readable. Why do the authors repeat the WHO data? This is publicly available and easy to find. Instead, they should focus on their actual report. How was the introduction restructured? The authors added a long paragraph that makes the brief report more tedious and harder to read, which does not help the paper.

I don’t see any change or improvement in providing clarity on the chosen sensing technologies and assessment methodologies, which would enhance the readers' understanding of how data will be collected and analyzed in the study.

Comments on the Quality of English Language

The paper needs extensive editing. This paper is hard to read because the graphics overlap and cover each other. The revisions in red make it impossible to read.

Author Response

Response to Reviewers' Comments

We express our sincere appreciation for taking the time and effort to review our article. We greatly value your comments and suggestions, which have been very helpful in improving the quality and scientific contribution of our work.

We appreciate your thorough review and attention to every detail. Your critical and constructive remarks have helped strengthen our research and refine our findings. Your experience and expertise have been invaluable to the development of our work.

The authors thank all reviewers for their valuable and constructive feedback and review. We have applied the suggestions and learned a lot from your comments; the document was updated and wholly restructured as suggested by the reviewers. In addition, the entire document was reviewed by an expert in the English language.

Note: In the answers below, the revised manuscript refers to the PDF document highlighted in yellow.

Reviewer 1

Unfortunately, the authors have not updated Figure 2, which fails to show the actual significance because the error bars are almost identical. This is my strongest criticism of the paper and should be easily addressed as per my previous review comment. It is disappointing that this simple data presentation was not performed to remove a clear potential source of biased results.

Additionally, while the authors have added a meta-analysis as a reference, this does not clarify the scope of expectation bias in these data, particularly given the low number of participants. The addition of error significance only allows for assumptions of significance between groups, suggesting the effect of implementing a Pulmonary Telerehabilitation Program may or may not be effective in treating post-COVID-19 syndrome in young patients. One way to address these doubts could be to include the actual data points in the graphic.

Adding more information to the introduction does not help to make the brief report more readable. Why do the authors repeat the WHO data? This is publicly available and easy to find. Instead, they should focus on their actual report. How was the introduction restructured? The authors added a long paragraph that makes the brief report more tedious and harder to read, which does not help the paper.

I don’t see any change or improvement in providing clarity on the chosen sensing technologies and assessment methodologies, which would enhance the readers' understanding of how data will be collected and analyzed in the study.

The paper needs extensive editing. This paper is hard to read because the graphics overlap and cover each other. The revisions in red make it impossible to read.

Dear Reviewer,

We sincerely appreciate your comments and regret that some of your concerns were not addressed in the previous version of the manuscript.

Here are the actions we have taken in response to your comments:

Figure 2: We have revised and updated Figure 2 to ensure the error bars accurately reflect the results. The values were added to the bottom of Table 3 to provide a more precise representation and minimize the risk of bias in interpreting the results. In addition, we have included individual data values in the graph. We are confident that these changes will adequately address your primary concern.

Expectation bias and error analysis: We recognize the importance of addressing expectation bias, especially given the small sample size. We have added a more detailed discussion of this aspect in the limitations section of the study, explaining how it may have influenced the results and how these limitations have been handled in the analysis. In addition, we have reworded the interpretation of the results to reflect the significance between groups more accurately based on the updated data.

Introduction: We have restructured the introduction to eliminate unnecessary repetition of WHO data and to focus the content more on the specific objective of our study. The introduction is more concise and direct, improving the flow and readability of the report.

Clarity in screening technologies and assessment methodologies: In response to your comment, we have improved the description of the detection technologies and assessment methodologies used in the study. A more detailed explanation of how the data were collected and analyzed is now provided, which we believe will facilitate better understanding by readers.

Readability and graphical issues: We have fixed the overlapping problems in the graphs and removed the red revisions that made the document difficult to read. We made sure that the current version of the manuscript is clear, neat, and easy to follow.

We thank you again for your detailed comments, which have been instrumental in improving the quality of the manuscript. We look forward to any additional comments you may have.

Reviewer 2 Report

Comments and Suggestions for Authors

Abstract: Line 17: Delete ’’ knowledge about’’

Line 20: It can not be the objective and aimed to?

So delete aimed

Line 23: Clarify what EG do. Also, there are punctuation mistakes, that should be resolved.

Line 34-38: Delete repeated words.

Repetition in the words throughout the manuscript (these words were added in the revised version) also the keywords.

I could not read the introduction. It needs again to be more neat. It can not be published in this form!!

Comments on the Quality of English Language

MODERATE 

Author Response

Reviewer 2

Abstract: Line 17: Delete" knowledge about" Line 20: It can not be the objective and aimed to?

Dear reviewer,

We sincerely appreciate your comments and observations, which have been of great help in improving our manuscript.

Abstract:

Line 17: We have removed the phrase "knowledge about," as per your suggestion, to improve the clarity and accuracy of the text.

Line 20: We have adjusted the wording to avoid ambiguity in the phrase "It can not be the object," and it has now been reworded so that the phrase clearly expresses the study's objective.

So delete aimed Line 23: Clarify what EG do. Also, there are punctuation mistakes, that should be resolved.

Dear reviewer,

Thank you very much for your comments and suggestions. We have made the following corrections in response to your comments:

Line 23: We have removed the word "aimed" to improve the accuracy of the text.

Clarification of the experimental group (EG): We have revised the wording to clarify the activities the experimental group (EG) participants performed in greater detail. It is now clearly specified that EG participants completed twelve remote and asynchronous pulmonary telerehabilitation sessions, including diaphragmatic breathing and aerobic exercises.

Punctuation errors: We have carefully reviewed the manuscript to correct punctuation and ensure the text flows correctly.

We thank you again for your meticulous review, which has significantly improved our manuscript.

 Line 34-38: Delete repeated words.  

Repetition in the words throughout the manuscript (these words were added in the revised version) also the keywords.

I could not read the introduction. It needs again to be more neat. It can not be published in this form!!

Dear Reviewer,

We sincerely appreciate your detailed and constructive comments. We have taken your comments into account and have made the following adjustments to the manuscript:

Lines 34-38: We have removed repeated words to improve clarity and avoid redundancies.

We have thoroughly revised the manuscript and critical terms, eliminating any unnecessary repetition in the revised version.

We recognize the importance of a clear and well-structured introduction. We have rewritten and reorganized the introduction to make it more concise and direct, improving the coherence and readability of the text.

We thank you again for your valuable suggestions, which have been essential to improving the quality of our work.